# Impact-Driven Event Covariates for Time-Series Foundation Models

Elham Alipour [1]   Xiaoli Zhang [1]   Tom Boyang Jin [1]   Alex Moschos [1]

## Abstract

Pretrained time-series foundation models can condition on future-known covariates, but event covariates are often limited to binary indicators or sparse categorical labels. This is problematic for forecast-based anomaly detection: in financial transaction series, recurring holidays, promotions, and settlement cycles induce predictable shifts that event-unaware systems may flag as false positives. We propose impact-driven event covariates, learned from event-centered residual responses across heterogeneous series rather than from event identity alone. Supplying these covariates to zero-shot Chronos-2 on a synthetic benchmark calibrated to empirical financial-transaction signatures reduces WAPE from 0.2872 to 0.2265 and improves anomaly-detection F1 from 0.2573 to 0.4580 relative to a no-event Chronos baseline. The learned covariates also transfer to unseen accounts with limited history without retraining. The results suggest that event covariates are most useful when their temporal alignment matches the underlying business process, providing an effective interface between domain event structure and pretrained time-series foundation models.

## 1. Introduction

Pretrained time-series foundation models can produce forecasts with little or no task-specific retraining, and recent models can condition on future-known covariates (Ansari et al., 2024; 2025; Arango et al., 2025). However, the quality of the covariates supplied to these models remains a bottleneck. In many operational time-series settings, holidays, promotions, settlement cycles, and deadlines are known in advance but encoded only as binary indicators or sparse categorical labels. Such encodings mark that an event occurs, but not what response it should induce, which series should

be affected, or when the response should occur.

Financial transaction time series provide a concrete stress test for this covariate-design problem. Forecast-based anomaly detectors that ignore event context can produce false positives during predictable event-driven periods, while binary event indicators cannot distinguish, for example, an immediate receivables spike from delayed cash settlement. We therefore ask: how should known recurring events be represented when used as future-known covariates for pretrained structured time-series models?

We propose impact-driven event covariates, learned from event-centered residual responses across heterogeneous series and supplied to a zero-shot Chronos-2 forecasting model. Our contributions are: (i) an event-covariate learning pipeline that turns realized cross-series impact into transferable covariates; (ii) a controlled synthetic benchmark for event-aware financial forecasting and anomaly detection, calibrated to empirical transaction signatures while preserving fully synthetic labels; and (iii) empirical evidence that learned event covariates improve forecasting and anomaly detection over no-event, binary-event, and classical baselines, including transfer to unseen accounts with limited history. The profile-level results further show that temporal alignment matters: lag-aware covariates help delayed-settlement cash accounts, while same-day covariates are strongest for receivable and revenue accounts.

## 2. Impact-Driven Event Covariates

**Task.** We observe daily signed transaction amounts $y_{i,t}$ for account $i$ at day $t$, together with a future-known event calendar. The goal is to forecast account-level transaction distributions and flag anomalies as deviations from the forecast distribution. We evaluate on a synthetic benchmark spanning three years of daily data, 1,000 accounts, four financial account families, realistic event calendars, and injected anomaly labels. The account families capture distinct operational regimes: cash accounts have business-day concentration and delayed settlement, receivables and revenue accounts have stronger promotional responses, and accrual accounts concentrate around close windows.

**Residual response extraction.** For each account, we estimate a non-event baseline from observed series rather than

[1]Amazon, Vancouver, BC, Canada. Correspondence to: Elham Alipour <ealipour@amazon.com>.

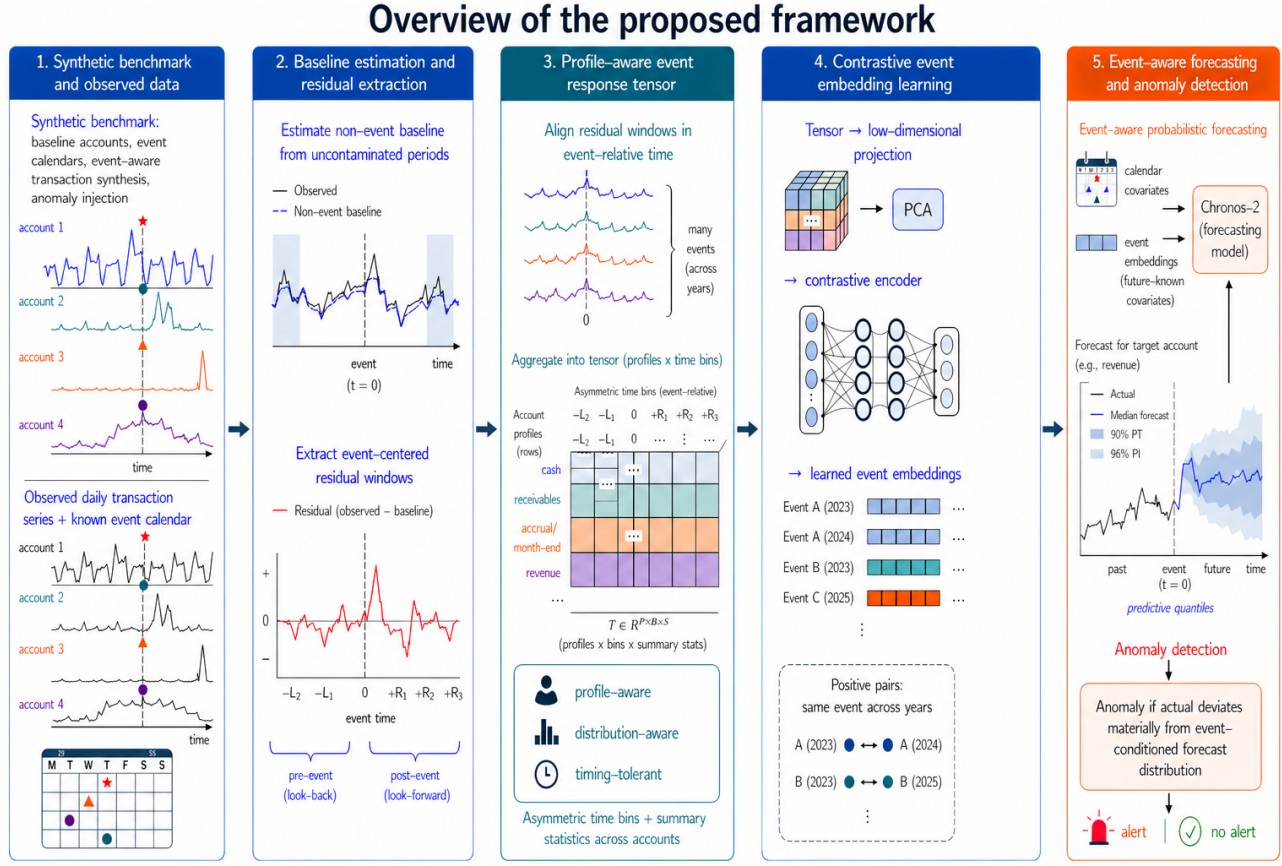

*Figure 1.* Overview of the pipeline. Event-centered residual responses are extracted from observed series, aggregated into profile-aware event-response tensors, transformed into learned event covariates, and supplied to a pretrained forecasting model for anomaly detection.

using simulator ground truth. Short windows around known events are excluded during baseline fitting to avoid leakage from the strongest event effects. Most accounts use weekday-specific rolling medians with month-end adjustments; month-end accounts use a separate close-window baseline. We compute robust normalized residuals $r_{i,t} = (y_{i,t} - \hat{b}_{i,t})/s_i$, where $\hat{b}_{i,t}$ is the estimated baseline and $s_i$ is a robust account-specific scale estimated on uncontaminated periods. These residuals are the only behavioral signal used to construct event representations.

**Event-response tensor.** For each event occurrence $e$, we align residuals in event-relative time within a window from 14 days before to 30 days after event start and aggregate them by account profile. Exact-day aggregation is brittle because event responses can shift slightly across accounts and years. We therefore use asymmetric relative-time bins: fine near the event core and coarser farther away. For each profile and time bin, we compute summaries including median response, spread, fractions of accounts with strong positive or negative residuals, and signed positive and negative response mass. We also append event-level strength and duration features such as duration, global response strength,

active profiles, and peak absolute response. The result is a tensor $T_e$ describing how event occurrence $e$ manifests across heterogeneous structured time series.

**Contrastive event embedding.** The tensor is standardized, reduced with PCA, and passed through a multilayer perceptron to produce a 16-dimensional event embedding $z_e = f_\theta(T_e)$, chosen as a compact representation after PCA indicated that the first 16 components retained about 75% of event-response variance. Positive pairs are occurrences of the same named event across different years, while negatives are other event occurrences in the batch. Following the InfoNCE contrastive objective (van den Oord et al., 2018), we define $s(e, e') = \exp(\mathrm{sim}(z_e, z_{e'})/\tau)$ and minimize

$$\mathcal{L}_e = -\log \frac{s(e, e^+)}{s(e, e^+) + \sum_{e^- \in \mathcal{N}(e)} s(e, e^-)}.$$

Here $\mathrm{sim}(\cdot, \cdot)$ denotes cosine similarity, $\tau$ is a temperature parameter, and $\mathcal{N}(e)$ is the set of in-batch negative event occurrences. We use the symmetric version of this loss, treating both members of each positive pair as anchors. This encourages recurring realizations of the same event to be close in embedding space while suppressing year-specific

noise and account-population variation. Intrinsic nearest-neighbor retrieval confirms that the representation preserves recurring event structure: the rich tensor with contrastive training achieves a 98.33% top-5 hit rate, compared with 70.00% for a simple tensor with PCA.

**Forecasting interface.** The learned representations are supplied to Chronos-2 as future-known covariates. We compare three temporal alignments: same-day, which attaches $z_e$ only to the event date; lag-bucketed, which reuses $z_e$ over event-relative lag buckets to capture delayed settlement; and phase-aware, which uses separate covariates for pre-, during-, early-post-, and late-post-event phases. These variants test whether the usefulness of an event representation depends on its alignment with the underlying business process. To avoid leakage, forecast-period events use fixed event-type embeddings obtained by averaging previous annual occurrences of the same recurring event; their own realized residuals are never used. Additional benchmark and embedding details are provided in Appendices A and B.

## 3. Experiments

**Setup.** For each forecast month in 2025, models receive a 24-month context window ending immediately before the forecast month. Chronos-based models output predictive quantiles at 0.02, 0.05, 0.10, 0.50, 0.90, 0.95, and 0.98; the median is used for point forecasts and the quantile pairs define 80%, 90%, and 96% prediction intervals. We compare the learned-covariate Chronos variants against no-event and binary-event Chronos baselines, Prophet with and without a binary event regressor (Taylor & Letham, 2018), Seasonal Naïve (Hyndman & Athanasopoulos, 2021), AutoETS (Hyndman et al., 2002), and AutoTheta (Assimakopoulos & Nikolopoulos, 2000). We report weighted absolute percentage error (WAPE) and interval score as the main forecasting metrics. Because downstream detection is based on prediction-interval breaches, interval score is especially relevant: it rewards narrow intervals only when they remain well calibrated. For anomaly detection, the final detector starts from days outside the 96% interval, filters by materiality relative to recent account scale, and applies duration-aware logic so that isolated deviations require stronger evidence than sustained runs. Detector ablations in Appendix D compare five operating points; adding materiality and duration filtering improves F1 substantially over raw interval breaches for all Chronos-based models, so we report this setting as the main detector.

**Forecasting.** Table 1 shows that learned event covariates improve over both no-event and binary-event Chronos variants. Lag-bucketed embeddings achieve the best WAPE, while phase-aware embeddings achieve the best interval score, suggesting that event representations improve both

*Table 1.* Forecasting comparison. Lower is better for both metrics. Prophet uses a 95% interval as an approximation to the 96% interval used by other models.

| Model | WAPE | Interval score |
|---|---|---|
| Lag-bucketed embedding | **0.2265** | 151,747 |
| Same-day embedding | 0.2308 | 145,645 |
| Phase-aware embedding | 0.2326 | **134,156** |
| Binary event | 0.2331 | 163,863 |
| No event (Chronos) | 0.2872 | 201,860 |
| Prophet + binary event | 0.3779 | 340,216 |
| AutoTheta (best classical) | 0.5507 | 437,527 |

*Table 2.* Aggregate anomaly detection under the final materiality- and duration-aware detector.

| Model | Precision | Recall | F1 |
|---|---|---|---|
| Lag-bucketed embedding | **0.3513** | 0.6576 | **0.4580** |
| Phase-aware embedding | 0.3110 | **0.7217** | 0.4347 |
| Binary event | 0.2984 | 0.6221 | 0.4034 |
| Same-day embedding | 0.2652 | 0.7121 | 0.3865 |
| No event (Chronos) | 0.1659 | 0.5730 | 0.2573 |
| Prophet + binary event | 0.0524 | 0.5880 | 0.0962 |
| AutoETS (best classical) | 0.0439 | 0.4025 | 0.0791 |

point accuracy and uncertainty calibration. Binary events also help relative to no-event Chronos, but they are consistently weaker than learned event covariates. Prophet and the classical baselines are substantially weaker in this setting, despite being deployment-relevant methods that can be fit independently per series. This suggests that both the pretrained backbone and learned event representation contribute to the gains.

**Anomaly detection.** Table 2 reports downstream anomaly detection using the same fixed detector across all models. The best aggregate F1 is obtained by lag-bucketed embeddings, followed by phase-aware embeddings. All event-aware Chronos variants outperform the no-event baseline, and the learned covariates outperform binary-event conditioning except for the same-day variant's aggregate F1. The improvement is practically important because the detector is intended as a screening-stage system: the goal is not to make final audit decisions automatically, but to improve the precision-recall tradeoff by suppressing predictable event-driven alerts while preserving recall.

**Where event covariates help.** The profile-level breakdown in Table 3 clarifies the mechanism. Lag-bucketed embeddings are best for cash accounts, consistent with delayed post-event settlement. Same-day embeddings are strongest for receivable and revenue accounts, where promotional responses concentrate near the event itself. Accrual accounts do not benefit materially from external event covariates, acting as a negative control. This pattern supports the interpretation that learned event covariates help when their

*Table 3.* Profile-level F1 for Chronos-based models under the final detector.

| Profile | Binary | Lag | No-event | Phase | Same-day |
|---|---|---|---|---|---|
| Accrual | 0.0862 | 0.0195 | **0.1075** | 0.0205 | 0.0355 |
| Cash | 0.3364 | **0.6052** | 0.2835 | 0.5956 | 0.2350 |
| Receivable | 0.5897 | 0.6062 | 0.3734 | 0.5682 | **0.6302** |
| Revenue | 0.3898 | 0.4311 | 0.1879 | 0.3758 | **0.5295** |

*Table 4.* Transfer to unseen accounts with limited history. Event embeddings are learned on a disjoint account set and reused without retraining. Lower WAPE and higher anomaly F1 are better.

| Model | WAPE | Anomaly F1 |
|---|---|---|
| Same-day embedding | **0.2458** | 0.3047 |
| Phase-aware embedding | 0.2499 | **0.3876** |
| Lag-bucketed embedding | 0.2502 | 0.3529 |
| Binary event | 0.2596 | 0.2534 |
| No event (Chronos) | 0.3233 | 0.1508 |

temporal structure matches the underlying business process, rather than uniformly improving all series.

**Transfer to unseen accounts.** To test whether the learned event covariates transfer beyond the accounts used during embedding construction, we evaluate them on a disjoint set of 1,000 accounts with only 13 months of history. The event embeddings are learned on a separate account set and reused without retraining. Table 4 shows that learned event covariates continue to outperform both binary-event and no-event baselines. This suggests that the embeddings capture transferable event-response structure rather than memorizing account-specific behavior.

## 4. Related Work

Our work connects three lines of research. First, pretrained time-series forecasting models such as Chronos and Chronos-2 enable zero-shot probabilistic forecasting, and recent extensions study how exogenous variables can be supplied to such models (Ansari et al., 2024; 2025; Arango et al., 2025). Related architectures also emphasize covariate-aware forecasting with exogenous variables (Wang et al., 2024; Yamaguchi et al., 2025). These methods generally treat covariates as given; our focus is how to construct informative event covariates from realized cross-series impact patterns. Second, time-series representation learning methods such as TS2Vec and CoST learn general-purpose sequence representations (Yue et al., 2021; Woo et al., 2022). In contrast, we learn representations of recurring event-response signatures, using cross-year event recurrence to define positive pairs. Third, financial and accounting anomaly-detection work has studied unsupervised and representation-based detectors (Thiprungsri & Vasarhelyi, 2011; Schreyer et al., 2019; Kiefer & Pesch, 2021), but typically treats

anomalies as context-free deviations. We instead define anomalies relative to an event-conditioned predictive distribution.

## 5. Discussion and Limitations

A pretrained forecaster can only condition on the information it receives. In our setting, the covariate interface is a practical bottleneck: replacing binary event indicators with impact-derived representations improves both forecast quality and anomaly detection. More generally, our results suggest that representation learning can be useful not only for modeling time series directly, but also for constructing the structured inputs supplied to pretrained forecasting models.

The temporal-alignment results further show that a single event representation is not always sufficient. The same event can be useful in different event-relative positions depending on the series family: lag-aware covariates help delayed-settlement cash accounts, while same-day covariates are strongest for receivable and revenue accounts. This suggests that effective event covariates should encode not only event identity or impact magnitude, but also how the event's effect aligns with the underlying operational process.

The study has limitations. The quantitative experiments use a synthetic benchmark rather than labeled production data; although the simulator was calibrated against empirical signatures from production financial-transaction and product-level data, synthetic benchmarks may omit real-world phenomena such as regime shifts, noisy metadata, changing business processes, incomplete event calendars, or unmodeled operational dependencies. Policy constraints prevent us from reporting internal production results, but parallel internal benchmarking showed comparable qualitative improvements, supporting the practical relevance of the synthetic findings. The benchmark is instantiated in a U.S. retail and financial-operations setting and assumes a reasonably clean, future-known event calendar, motivating future work on robustness across noisier calendars, regions, industries, and real analyst-review outcomes.

## 6. Conclusion

We introduced impact-driven event covariates for pretrained structured time-series foundation models. By learning event representations from event-centered residual responses and using them as future-known covariates, the method improves forecasting and forecast-based anomaly detection over no-event and binary-event baselines, including transfer to unseen accounts with limited history. The results show that temporal alignment matters and suggest that learned covariate interfaces are an important complement to advances in foundation forecasting backbones.

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

## A. Synthetic Benchmark Details

The benchmark spans three years of daily data (2023–2025) and contains 1,000 accounts drawn from four behavioral families: cash/banking, receivables, accrual/month-end, and revenue. Accounts are generated with profile-specific activity calendars, weekly and monthly structure, transaction-count dispersion, amount scales, sign policies, and trend parameters. Daily counts are sampled from a negative-binomial model and transaction magnitudes from a lognormal model.

The simulator was calibrated against empirical summaries and qualitative response signatures from production transaction data. These signatures were used to anchor weekday activity patterns, month-end concentration, sparsity, transaction-scale ranges, settlement lags, return timing, and event-response magnitudes across the four account families. The goal of this calibration is not to claim that the benchmark fully reproduces a production ledger, but to ensure that the synthetic setting reflects realistic statistical and temporal regimes while retaining fully synthetic labels.

The event layer combines deterministic calendar rules with constrained LLM-generated event structure. Known recurring events such as Thanksgiving, Black Friday, Cyber Monday, and Memorial Day are anchored with deterministic date logic. The LLM-generated portion produces structured yearly event lists and account-type-specific impact profiles under schema constraints, bounded multiplier ranges, and validation checks. The LLM does not directly emit time series; instead, impact profiles are converted into day-level multipliers by a statistical engine that supports pre-event ramps, during-event effects, settlement waves, returns, and month-end cascades. Overlapping events are combined with a dominant-event-wins rule and hard caps. The purpose of the LLM component is to provide semantic variety in event types and plausible account-type-specific impact profiles; numerical effects remain constrained by schemas, bounded ranges, deterministic date corrections, and downstream statistical sampling. Each account is also assigned latent sensitivity factors sampled from profile-specific Beta distributions, governing its response to holidays, promotions, and event magnitude. These factors ensure that accounts within the same broad family respond heterogeneously to the same event, while preserving family-level response patterns such as delayed settlement in cash accounts and near-event promotional spikes in receivable and revenue accounts.

Anomalies are injected only after clean event-aware series have been generated. In the default configuration, 30% of accounts receive anomalies, each affected account receives 1–3 anomalies per year, and severity is sampled across minor, moderate, and major tiers. The anomaly registry includes magnitude spikes, magnitude drops, sign reversals, gradual drifts, missing month-end close behavior, and missing expected event response. Each injected anomaly is exported with account, date range, severity, event context, anomaly type, and raw injection parameters.

## B. Additional Embedding Details

Event-response windows span 14 days before event start to 30 days after event start. Event-relative time is grouped into asymmetric bins that are narrower near the event core and wider in the tails. For each event occurrence, account profile, and time bin, the tensor includes distributional summaries such as median response, spread, quantiles, fractions of strongly positive or negative residuals, and signed positive and negative mass. Additional temporal-shape descriptors summarize peak response, total response mass, temporal center and spread, and region-level summaries over pre-event, event-core, and post-event phases. Event-level strength features include duration, global response strength, impact per day, number of active profiles, and peak absolute response.

The final tensor is standardized and reduced with PCA before contrastive training. The encoder is a multilayer perceptron with a 16-dimensional output embedding. We used 16 dimensions as a compact default: exploratory PCA indicated that the first 16 components retained about 75% of the variance in the event-response features, suggesting a reasonable tradeoff between compression and representational capacity. Training uses positive pairs formed by same-name event occurrences across different years, a symmetric InfoNCE objective with temperature 0.15, batch size 32, learning rate $10^{-3}$, weight decay $10^{-4}$, and early stopping.

For the intrinsic evaluation, we compare two tensor constructions. The *simple tensor* aggregates residual responses by account profile and exact relative day using only coarse central-tendency summaries. The *rich tensor* is the final representation described above: it uses asymmetric event-relative time bins, distributional summaries, temporal-shape descriptors, and event-level strength features. The comparison isolates whether the improvement comes from the richer event-response representation itself or only from the downstream embedding model. Table 5 reports intrinsic nearest-neighbor retrieval results for the event representations. The progression from a simple tensor to the richer tensor shows that preserving profile-specific, distributional, and timing-tolerant response structure is important; applying contrastive training to the richer tensor further improves same-event grouping.

*Table 5.* Intrinsic evaluation of event representations.

| Representation / model | Avg. same-name hits in top-5 | Top-5 hit rate |
|---|---|---|
| Simple tensor + PCA | 1.0166 | 70.00% |
| Simple tensor + autoencoder | 1.0833 | 76.67% |
| Rich tensor + PCA | 1.4167 | 90.00% |
| Rich tensor + contrastive encoder | 1.9333 | 98.33% |

## C. Additional Forecasting and Transfer Results

### C.1. Extended forecasting metrics

Table 6 reports the full forecasting metrics. The main paper focuses on WAPE and interval score because they align most directly with forecast-based anomaly detection: WAPE summarizes point forecast quality, while interval score rewards prediction intervals that are both narrow and well calibrated. The broader metrics show the same qualitative pattern, with learned event covariates outperforming no-event, binary-event, Prophet, and classical baselines.

*Table 6.* Extended forecasting metrics. Lower is better for MAE, RMSE, WAPE, and interval score. Coverage is reported at the nominal 96% level for Chronos-based models and 95% for Prophet.

| Model | MAE | RMSE | WAPE | Coverage | Interval score |
|---|---|---|---|---|---|
| Lag-bucketed embedding | 14,174.19 | 51,892.39 | 0.2265 | 0.9856 | 151,747 |
| Same-day embedding | 14,656.26 | 53,644.57 | 0.2308 | 0.9837 | 145,645 |
| Phase-aware embedding | 14,669.67 | 54,909.43 | 0.2326 | 0.9849 | 134,156 |
| Binary event | 15,059.53 | 58,962.80 | 0.2331 | 0.9807 | 163,863 |
| No event (Chronos) | 18,382.05 | 72,165.44 | 0.2872 | 0.9747 | 201,860 |
| Prophet + binary event | 23,858.36 | 87,884.14 | 0.3779 | 0.9625 | 340,216 |
| Prophet | 25,403.03 | 92,414.57 | 0.3995 | 0.9658 | 362,271 |
| AutoTheta | 34,909.20 | 116,259.02 | 0.5507 | 0.9719 | 437,527 |
| AutoETS | 37,583.74 | 142,917.32 | 0.5851 | 0.9729 | 507,381 |
| Seasonal Naïve | 38,370.59 | 146,488.00 | 0.6053 | 0.9641 | 693,231 |

### C.2. Transfer to unseen accounts with limited history

To test whether event representations transfer beyond the accounts used during embedding construction, we ran an additional experiment on a disjoint set of 1,000 accounts that were not used to learn the event embeddings and that had only 13 months of training history available. This setting approximates newly added accounts in production, where long account-specific histories are often unavailable. The evaluation accounts shared the same event environment as the main benchmark, but the embeddings themselves were learned on a separate account set and reused without retraining.

Table 7 reports forecasting performance in this reduced-history transfer setting. The qualitative pattern remains consistent with the main experiments: all learned event-covariate variants outperform both the binary-event and no-event baselines. Same-day embeddings achieve the best WAPE, while phase-aware embeddings achieve the best interval score.

*Table 7.* Forecasting on unseen accounts with limited history. Event embeddings are learned on a disjoint account set and reused without retraining. Lower is better.

| Model | WAPE | Interval score |
|---|---|---|
| Same-day embedding | **0.2458** | 190,127 |
| Phase-aware embedding | 0.2499 | **167,443** |
| Lag-bucketed embedding | 0.2502 | 203,115 |
| Binary event | 0.2596 | 229,981 |
| No event (Chronos) | 0.3233 | 289,553 |

Table 8 reports downstream anomaly-detection performance under the same transfer setting. Although absolute anomaly performance is lower than in the main experiment, the relative advantage of learned event covariates remains strong. All

three learned variants outperform both binary-event and no-event baselines on F1, suggesting that the learned representations capture transferable event-response structure rather than memorizing the accounts used during embedding learning. The transfer setting also changes the relative ranking of temporal alignments. While lag-bucketed embeddings achieve the strongest aggregate anomaly F1 in the main experiment, phase-aware embeddings perform best on unseen accounts with limited history. We interpret this as a robustness effect: lag-bucketed alignment is especially effective when sufficient account history is available to support delayed-settlement dynamics, whereas phase-aware covariates provide a coarser event-lifecycle structure that transfers better when account-specific history is limited.

*Table 8.* Aggregate anomaly detection on unseen accounts with limited history under the final detector. Event embeddings are learned on a disjoint account set and reused without retraining.

| Model | Precision | Recall | F1 |
|---|---|---|---|
| Phase-aware embedding | **0.2768** | 0.6480 | **0.3876** |
| Lag-bucketed embedding | 0.2509 | 0.5948 | 0.3529 |
| Same-day embedding | 0.1957 | **0.6876** | 0.3047 |
| Binary event | 0.1596 | 0.6139 | 0.2534 |
| No event (Chronos) | 0.0870 | 0.5648 | 0.1508 |

## D. Detector Ablation Summary

Table 9 reports the detector ablation across the Chronos-based forecasting models. We compare five increasingly selective operating points: raw 90% interval breaches, 90% breaches with materiality filtering, raw 96% interval breaches, 96% breaches with materiality filtering, and the final 96% materiality- and duration-aware detector used in the main text. The ablation shows a consistent precision–recall tradeoff. Raw interval breaches achieve high recall but produce many false positives, while adding materiality filtering substantially improves precision. Adding duration-aware logic further improves the operating point by suppressing isolated low-value breaches while retaining sustained deviations.

Across all Chronos-based models, the final detector achieves the strongest F1. For example, for the lag-bucketed embedding model, F1 increases from 0.1559 under raw 96% interval breaches to 0.2213 with materiality filtering and to 0.4580 after adding duration-aware logic. Similar improvements are observed for the no-event, binary-event, phase-aware, and same-day variants. This supports the use of the 96% prediction-interval breach detector with materiality and duration filtering as the main detector in the paper, since it provides a more practical screening-stage operating point than raw interval breaches, improving precision while preserving useful recall.

*Table 9.* Detector ablation across Chronos-based forecasting models. The final 96% materiality- and duration-aware detector achieves the strongest F1 for each model.

| Model | Detector | Precision | Recall | F1 |
|---|---|---|---|---|
| Binary event | Outside 90% PI | 0.0198 | 0.8840 | 0.0388 |
| Binary event | Outside 90% PI + materiality | 0.0464 | 0.8117 | 0.0878 |
| Binary event | Outside 96% PI | 0.0641 | 0.6617 | 0.1169 |
| Binary event | Outside 96% PI + materiality | 0.1079 | 0.6357 | 0.1845 |
| Binary event | Outside 96% PI + materiality + duration | 0.2984 | 0.6221 | **0.4034** |
| Lag-bucketed embedding | Outside 90% PI | 0.0181 | 0.8772 | 0.0355 |
| Lag-bucketed embedding | Outside 90% PI + materiality | 0.0424 | 0.8090 | 0.0806 |
| Lag-bucketed embedding | Outside 96% PI | 0.0878 | 0.6958 | 0.1559 |
| Lag-bucketed embedding | Outside 96% PI + materiality | 0.1323 | 0.6767 | 0.2213 |
| Lag-bucketed embedding | Outside 96% PI + materiality + duration | 0.3513 | 0.6576 | **0.4580** |
| No event | Outside 90% PI | 0.0152 | 0.8158 | 0.0299 |
| No event | Outside 90% PI + materiality | 0.0300 | 0.7667 | 0.0577 |
| No event | Outside 96% PI | 0.0466 | 0.6180 | 0.0866 |
| No event | Outside 96% PI + materiality | 0.0714 | 0.5948 | 0.1274 |
| No event | Outside 96% PI + materiality + duration | 0.1659 | 0.5730 | **0.2573** |
| Phase-aware embedding | Outside 90% PI | 0.0183 | 0.8936 | 0.0360 |
| Phase-aware embedding | Outside 90% PI + materiality | 0.0400 | 0.8131 | 0.0763 |
| Phase-aware embedding | Outside 96% PI | 0.0901 | 0.7572 | 0.1610 |
| Phase-aware embedding | Outside 96% PI + materiality | 0.1315 | 0.7326 | 0.2230 |
| Phase-aware embedding | Outside 96% PI + materiality + duration | 0.3110 | 0.7217 | **0.4347** |
| Same-day embedding | Outside 90% PI | 0.0188 | 0.9127 | 0.0368 |
| Same-day embedding | Outside 90% PI + materiality | 0.0430 | 0.8295 | 0.0817 |
| Same-day embedding | Outside 96% PI | 0.0854 | 0.7626 | 0.1536 |
| Same-day embedding | Outside 96% PI + materiality | 0.1240 | 0.7312 | 0.2120 |
| Same-day embedding | Outside 96% PI + materiality + duration | 0.2652 | 0.7121 | **0.3865** |

