# OpenReview forum: "Impact-Driven Event Covariates for Time-Series Foundation Models"
_ICML.cc/2026/Workshop/FMSD — FMSD @ ICML 2026 Poster_

### Official Review · Reviewer_A18f · 2026-05-14
**Promising Event-Covariate Design with Evaluation and Ablation Gaps**

**Rating:** 7
**Confidence:** 3

**Review:**

### Summary

This paper proposes impact-driven event covariates for time-series foundation models. Instead of representing known events as binary indicators, the authors construct event-response tensors from residual behaviour around events, learn compact embeddings with a contrastive objective, and provide these embeddings as future-known covariates to Chronos-2 for forecasting and anomaly detection. The method is evaluated on a synthetic financial-transaction benchmark and shows improvements over no-event, binary-event, Prophet, and classical forecasting baselines.

### Strengths

The paper addresses a relevant and practical problem: forecast-based anomaly detectors can produce false positives around predictable events such as holidays, promotions, settlement cycles, and month-end processes. The idea of learning event covariates from realised impact rather than event identity alone is interesting and well suited to the workshop’s focus on foundation models for structured time-series data. The empirical results are promising, especially the improvements over no-event Chronos and the profile-level analysis showing that different temporal alignments help different account types. The transfer experiment on unseen accounts is also a useful addition.

### Areas for Improvement

The main limitation is that the evaluation is entirely synthetic. The authors acknowledge this, but the simulator is central to the paper and is only described at a high level, even in the appendix. Since the simulator defines the event dynamics, anomaly types, and labels, more detail is needed to judge whether the method generalises beyond the data-generating process. At least one real-world or semi-real-world benchmark, such as M5 or Corporación Favorita, would strengthen the claims.

A second concern is that the current comparison may conflate representation quality with temporal alignment. The learned embeddings are evaluated with same-day, lag-bucketed, and phase-aware variants, while the binary baseline appears less temporally expressive. The same-day learned embedding is only marginally better than binary events for WAPE, and binary events perform better than same-day embeddings on aggregate anomaly F1. This suggests that some gains may come from phasing/lag structure rather than the learned embedding itself.

The temporal protocol for constructing event-response tensors should also be clarified. Since the tensors are built from realised residual responses around events, it is important to know how covariates are produced for forecast-period events whose responses are not yet observed. For example, are embeddings constructed from historical occurrences of each recurring event type and then reused for future occurrences, or are occurrence-specific tensors built for each event? The former would be a valid and useful setting for recurring known events, while the latter could introduce target leakage if evaluation-period residuals are used. Clarifying this would also help define the scope of the method, which appears most naturally suited to recurring events with historical response data.

### Detailed Comments
- Please include at least one real-world or semi-real-world benchmark for the forecasting results, such as M5 or Corporación Favorita. Even without anomaly labels, these event-rich retail datasets would help test whether the learned event covariates generalise beyond the synthetic simulator.
- Please clarify the temporal protocol for constructing the event response tensors. Are tensors for evaluation-period events built only from data available before the forecast origin?
- The paper appears to construct event-response tensors per event occurrence, using residuals in a window around that occurrence. This raises an important deployment and evaluation question: how are covariates produced for forecast-period event occurrences whose responses are not yet observed? If occurrence-specific tensors are constructed using evaluation-period residuals, this would introduce target leakage. If instead historical occurrence-level tensors are aggregated into an event-type prototype and reused for future events, this should be stated explicitly.
- Please provide more detail on the simulator, including equations, parameter ranges, event-generation rules, and validation against empirical signatures.

### Justification of Score

I would give this paper: Good paper, accept. The problem is relevant, the proposed direction is interesting, and the results are promising for a workshop setting. However, the fully synthetic evaluation, unclear temporal protocol in constructing event response tensors, and missing ablations around temporal alignment weaken the strength of the claims. The paper is a good fit for discussion at the workshop, but the main empirical conclusions should be interpreted cautiously until these issues are addressed.

---

### Official Review · Reviewer_zTuE · 2026-05-15
**Promising ideas to include known events as covariates, but lacks proper evaluation and readability**

**Rating:** 4
**Confidence:** 4

**Review:**

**Summary**

The paper aims to improve forecasting and anomaly detection by providing pretrained foundation models with event-aware covariates. The authors propose a method to learn the representation of known events, which outperforms usual binary indicators.

The paper focuses on anomaly detection based on observed deviations from a baseline predictor. Without any event information, these pipelines can return many false positives. Yet, practitioners usually know in advance of upcoming events (holidays, deadlines, etc.). The authors propose to encode the known events using contrastive learning.

To do so, synthetic accounts with simulated events were generated, and event representations are build by: extracting residuals of the baseline predictor on windows surrounding the realized events, computing various statistical features, applying a PCA and then a neural network. The network is trained via a contrastive objective. This representation brings recurring identical events together and moves different events away from each other. This largely improves same-event retrieval, as shown by the authors.

This representation can be fed as a covariate to pretrained forecasting models, either: specifically at the event time-stamp ("same-day"), on the window surrounding the event ("lag-bucketed"), or with different values within the window ("phase-aware"). Results show that the event-covariates improve models both on forecasting and anomaly detection. The choice of the embedding strategy depends on the type of account.

**Strengths**

The authors tackle an important topic, which is the inclusion of sparse events as covariates in forecasting models. Their learned representation improves upon no-context and naive binary strategies on their synthetic dataset. The multiple strategies to align the obtained representations depending on the type of account is a good idea.

**Areas for Improvement**

The paper is very textually dense and uses many different keywords, without defining them properly. For example, post-processing operations such as "materiality filtering" and "duration-aware logic" are not detailed, yet they are a crucial part of the pipeline. The overall pipeline can be explained more concisely but also more clearly (figure 1 is not very informative, and isn't cited in the paper).

More importantly, the evaluation of the overall method is not very convincing. It is only conducted on a single synthetic dataset (of which the generation pipeline is not very detailed), and lacks naive event-included baselines.

**Detailed Comments**

The evaluation of the method should include more important baselines than simple binary encoding. Has no related works dealth with this problem? This paper itself introduces baselines in the ablation study which could be included in the forecasting and anomaly detection experiments (the ablations were only conducted with respect to the same-event retrieval metric). Details on the evaluation procedure should also be included in appendix: what temporal splits were conducted for training the representation and evaluating the models? How exactly are values flagged as anomalies?

I would also advise the author to introduce the task and pipeline more precisely, perhaps with more formulas and less abstract keywords. In general, any new keyword should be defined somewhere. This includes important novel keywords ("materiality", "duration-aware logic"), implicit keywords prone to interpretation ("month-end adjustments", "event-centered", "impact-driven", "materiality", "non-event") as well metrics from the literature which could be added to appendices ("interval score", "hit-rate").

To better grasp the whole pipeline, more precise details should be provided on the different types of accounts that were generate, and the features that are extracted to build representation of events. At least in the form of concise tables. Currently, appendix A and B do not provide much more information than the main text.

**Justification of score**

The paper proposes a promising method to introduce known events in the inference pipeline, but lacks proper evaluation. The paper is not easy to read because many terms and the steps of the pipeline are introduced without precise definitions.

---

### Official Review · Reviewer_SRN4 · 2026-05-20
**Useful event-covariate idea, but validation relies too heavily on a synthetic benchmark**

**Rating:** 5
**Confidence:** 4

**Review:**

**Summary**

This paper proposes impact-driven event covariates for pretrained time-series foundation models. Instead of representing known events only through binary indicators or sparse event labels, the method learns event embeddings from event-centered residual responses across heterogeneous financial transaction series. These learned covariates are then supplied to a zero-shot Chronos-2 forecasting model for forecasting and forecast-based anomaly detection.

The paper evaluates the approach on a synthetic financial-transaction benchmark with 1,000 accounts, multiple account families, realistic event calendars, and injected anomaly labels. The learned event covariates improve both forecasting and anomaly detection compared with no-event and binary-event Chronos baselines. The paper also studies different temporal alignments, including same-day, lag-bucketed, and phase-aware covariates.

Overall, the idea is useful and clearly motivated. However, the evidence is not fully convincing because the main evaluation is synthetic, the event-representation baselines are relatively simple, and some possible leakage issues in event-embedding construction need clearer treatment.

**Strengths**

The paper addresses an important practical problem: forecast-based anomaly detection can produce false positives around predictable event-driven periods, and binary event indicators are often too weak to represent event impact.

The proposed event representation is intuitive and well aligned with the problem. Learning event covariates from residual responses captures not just event identity, but also the magnitude, timing, and profile-specific structure of event impact.

The results are directionally strong. Learned covariates improve WAPE, interval score, and anomaly-detection F1 over no-event and binary-event Chronos baselines.

The transfer experiment on unseen accounts with limited history is a valuable addition, since it suggests that the learned event covariates capture reusable event-response structure rather than only account-specific behavior.

**Areas for Improvement**

The main limitation is that the quantitative evaluation is entirely synthetic. Although the benchmark is calibrated to empirical financial-transaction signatures, the simulator defines the event effects and the anomaly-generation process. This makes it difficult to know how well the method would perform with noisy, incomplete, shifting real-world event calendars.

The event-representation baselines are not strong enough. The paper compares against no-event and binary-event Chronos baselines, but stronger event-feature baselines would make the contribution clearer. For example, one-hot event identity embeddings, hand-crafted event-lag features, event-type plus phase features, or supervised event embeddings would be useful comparisons.

Potential leakage in event-embedding construction should be clarified. Since event covariates are learned from realized residual responses, the paper should clearly state whether embeddings are constructed only from training-period data, or whether any future/test-period responses influence the covariates used for forecasting.

The anomaly-detection results depend strongly on detector engineering. The final detector uses interval breaches, materiality filtering, and duration-aware logic. This is practical, but the paper should more clearly separate improvements due to better event covariates from improvements due to detector post-processing.


**Detailed Comments**


Please clarify the temporal construction of event embeddings. Are event-response tensors learned only from training data available before the forecast period? If not, the method may indirectly use future information.

Please evaluate on at least a small real or semi-real benchmark if possible. Even anonymized aggregate production results, or a partially real event-calendar evaluation, would substantially strengthen the paper.

Please clarify whether the contrastive event embeddings are learned separately for each experiment or fixed once. This matters for interpreting transfer to unseen accounts and limited-history settings.

Please clarify whether Chronos-2 was used strictly zero-shot or whether any adaptation was performed. This is important because the paper’s main framing depends on using learned covariates as an interface to a pretrained model.


**Justification of Score**

I would assign this paper a 5: marginally below acceptance threshold.

The paper is clear, relevant, and technically reasonable. The proposed event-covariate learning pipeline is useful, and the results show consistent gains over no-event and binary-event baselines. The temporal-alignment and transfer analyses are also valuable.

However, the current evidence is not strong enough for a higher score. The evaluation is synthetic-only, stronger event-representation baselines are missing, leakage controls need clearer explanation, and the anomaly-detection gains partly depend on detector engineering. Overall, I view the paper as a promising contribution, but it needs stronger validation and more transparent experimental controls before the claims are fully convincing.